# Using Machine Learning to Discover Latent Social Phenotypes in Free-Ranging Macaques

**DOI:** 10.3390/brainsci7070091

**Published:** 2017-07-21

**Authors:** Seth Madlon-Kay, Lauren J. N. Brent, Michael J. Montague, Katherine A. Heller, Michael L. Platt

**Affiliations:** 1Department of Neuroscience, Perelman School of Medicine, University of Pennsylvania, Philadelphia, PA 19104, USA; montag@mail.med.upenn.edu (M.J.M.); mplatt@mail.med.upenn.edu (M.L.P.); 2Center for Research in Animal Behaviour, University of Exeter, Exeter EX4 4QG, UK; L.J.N.Brent@exeter.ac.uk; 3Department of Statistical Science, Duke University, Durham, NC 27708, USA; kheller@stat.duke.edu; 4Department of Psychology, School of Arts and Sciences, University of Pennsylvania, Philadelphia, PA 19104, USA; 5Marketing Department, The Wharton School, University of Pennsylvania, Philadelphia, PA 19104, USA

**Keywords:** machine learning, behavioral genetics, social neuroscience, animal models

## Abstract

Investigating the biological bases of social phenotypes is challenging because social behavior is both high-dimensional and richly structured, and biological factors are more likely to influence complex patterns of behavior rather than any single behavior in isolation. The space of all possible patterns of interactions among behaviors is too large to investigate using conventional statistical methods. In order to quantitatively define social phenotypes from natural behavior, we developed a machine learning model to identify and measure patterns of behavior in naturalistic observational data, as well as their relationships to biological, environmental, and demographic sources of variation. We applied this model to extensive observations of natural behavior in free-ranging rhesus macaques, and identified behavioral states that appeared to capture periods of social isolation, competition over food, conflicts among groups, and affiliative coexistence. Phenotypes, represented as the rate of being in each state for a particular animal, were strongly and broadly influenced by dominance rank, sex, and social group membership. We also identified two states for which variation in rates had a substantial genetic component. We discuss how this model can be extended to identify the contributions to social phenotypes of particular genetic pathways.

## 1. Introduction

Understanding how biological and environmental factors shape human social phenotypes—the ways in which we interact with others and in which others shape our own behavior—is a topic of intense interest in neuroscience. While neuroscientific research has made impressive strides in identifying genetic pathways and brain areas that contribute to specific social phenotypes, traditional approaches are limited in their ability to assess how biological and environmental factors contribute to naturalistic social behaviors in real-world environments. Animals used in laboratory research may differ from humans in the brain areas associated with social cognition, and live in laboratory environments with much less varied and dynamic social interactions than natural environments [1]. Research in humans faces similar limitations, as comprehensive, in-depth observations of human social behavior in natural environments is both logistically and ethically problematic, while controlled laboratory tasks take place in restricted and low-stakes environments which may leave the true social environment unobserved.

An alternative approach to understanding biological contributions to human social behavior is to study free-ranging animals whose biology and social behavior more closely approximate our own. Free-ranging non-human primates (NHPs) provide naturalistic and human-relevant variability in social behavior that cannot be modeled in the laboratory or easily (or ethically) collected in humans. With free-ranging NHPs, it is possible to collect comprehensive data on many different aspects of social behavior, as well as both noninvasive and invasive biological samples and life history information that can be used to disentangle genetic and environmental influences. However, natural social behavior is both noisy and high-dimensional, presenting severe challenges for traditional methods of data analysis. Standard methods for relating genetic variation to traits take an atomistic view by measuring genetic influences on each measured variable individually. While this approach makes sense for specific, well-defined phenotypes of interest such as educational attainment [2] or height [3], it is less useful for high-dimensional data where any individual dimension is of limited or unknown significance.

In the case of natural behavior, continuous streams of activity are necessarily segmented into many discrete behavioral categories during measurement, in many cases with no specific behavior being of special interest relative to the whole. The challenge is thus to develop a rigorous and biologically meaningful approach for reconstructing a “whole” behavioral state based on discrete, atomized behaviors. Such a challenge can be viewed as a type of latent structure learning, in that there is some underlying structure that is not directly observed but that ties together many different individual pieces of data. Models of this sort are increasingly popular in the field of probabilistic machine learning [4]. Though machine learning models are most frequently used for prediction, here we adapt methods from the machine learning literature to infer a meaningful and useful structure in observational data sets.

Any source of genetic variation is unlikely to affect only a single discrete social behavior. A genetic variant associated with grooming behavior, for instance, may also be associated with increased time spent in the proximity of other animals in general. Genetic influences are more likely distributed across many different individual social behaviors than localized to specific types of actions [5]. Furthermore, in the case of observations of natural social behavior, it is important to recognize that each type of behavior does not occur in isolation. Rather, the set of behaviors that occur during a given observational session may reflect the current situation or behavioral state the animal is in. The relationship between any given behavior and an underlying social phenotype thus may vary with context. For example, adjusting a watchband while speaking to a group of people may be a nervous tick reflecting social anxiety, while adjusting a watchband in one’s bedroom might reflect only an ill-fitting watchband.

Thus, it is critical for any useful model of observed social behavior to reflect the complex interrelatedness of individual behaviors, while retaining identifiability and scalability to large data sets with dozens of individual behaviors and thousands of observations. To achieve this goal, we here adapt a family of models from machine learning known as topic models. Topic models are tools for describing the topics that are covered by a corpus of documents and to what degree each document is focused on each topic [6,7]. In our analysis, each animal is analogous to a document, while the animal’s social phenotype, or the weights assigned to different behavioral states, correspond to the weights a document assigns to each of the different topics. Under a topic model, each word in a document belongs to a topic, just as in the current model each observational session belongs to a behavioral state. Of the many topic modeling variants that exist in the literature, the current model is most closely related to the logistic normal topic model [8,9]. Our model differs from the standard logistic normal topic model in that we do not explicitly model correlations between topic weights (here, weights on behavioral states); instead, our model uses a hierarchical regression layer to incorporate outside information about documents (here, animals) in predicting their phenotypes. To the best of our knowledge, no previous topic model has included hierarchical regression at the level of topic weights (but see [10] for a related model).

Recent research on behavioral phenotypes in NHPs has used principle components analysis (PCA) to identify latent structure in natural behavior [5,11], as well as in the study of animal behavior generally [12]. However, that research relied on rates of behaviors averaged across many different observational periods on the same animal. This means that the relationships among behaviors and the patterns in which they co-occur within a single observation are lost. That is, PCA captures variation among animals, but at the cost of losing all information on variability at the level of individual observational periods. By contrast, the hierarchical nature of topic models allows us to explicitly capture variability at both the level of individual animals and the level of observational sessions, in the form of the phenotypes used to represent individual animals and the behavioral states used to characterize observations, respectively.

Our behavioral data is a large set of animal observations taken from the free-ranging rhesus macaque (*Macaca mulatta*) colony on Cayo Santiago Island off the coast of Puerto Rico. Rhesus macaques provide an excellent model species for studying biological and environmental influences on human social behavior due to their extensive use in lab and field research; neural circuitry that is homologous to that of humans; and complex social behaviors that are critical to their biological success [13,14,15,16]. Because demographic, pedigree, and genetic data are also available for the colony [17,18], this detailed record of natural social behavior provides a unique opportunity for modeling genetic, demographic, and environmental influences on complex social phenotypes.

## 2. Materials and Methods

### 2.1. Study Site and Subjects

The studied population is a colony of rhesus macaques (*Macaca mulatta*) living on the island of Cayo Santiago, a 37-acre island 1 km off the southeastern coast of Puerto Rico. This is a free-ranging, freely-breeding population with known pedigrees, rich data on life histories and fitness, and extensive genetic and observational data on behavior. The colony was founded in 1938 with a population of 409 Indian-origin rhesus macaques and is currently maintained by the University of Puerto Rico Medical Sciences Campus [19]. The current population numbers approximately 1600 animals self-organized into six different social groups. Approximately 600 of the animals are adults of age six or above, and 600 are juveniles between the ages of one and five. Researcher and caretaker intervention in the population is minimal. Animals in the colony are provided commercially available monkey chow daily and unlimited access to water. Animals are handled only during designated annual trapping periods, during which infants are tagged for identification and population control can be implemented via the removal of small numbers of animals.

The data used in this study were 8205 ten-minute focal observations collected from adult (age > 6 years) male and female macaques from three social groups (F, HH, and R). Observational data was collected from group F during 2012 and 2013 and from group HH and R during years 2014 and 2015, respectively. A total of 227 macaques (71 male) were represented in this data set, with 106 from group F (41 male), 41 from group HH (11 male), and 80 from group R (19 male). The number of focal observations per animal ranged from 80 (approximately 13.33 h) to 11 (approximately 1.83 h), with an average of 36.15 focal observations per animal (approximately 6 h).

### 2.2. Observational Data

The data is comprised of ten-minute focal samples, wherein a trained researcher tracks an individual monkey (the focal animal) for a ten-minute period while comprehensively recording the animal’s behaviors [17,18]. Once each ten-minute session is complete, the observer moves on to a new focal animal. Observers recorded the times at which the monkey engaged in any behaviors specified by a predetermined ethogram, as well as the identities of nearby animals during the focal observation [17]. The order in which animals were observed was semi-randomized to equalize the times of day and year that animals were observed. The ethogram used to guide data collection consisted of both social and non-social behaviors, both of which were classified as either activities, which have durations, or events, which occur over effectively zero duration. Social behaviors were further classified into affiliative and agonistic behaviors. Each social behavior was further divided based on whether the behavior was initiated by another animal and directed towards the focal animal, or initiated by the focal animal towards another animal. A total of 30 behaviors were used in the present study, including behaviors divided by giving versus receiving. For a description of each behavior included in the model, see the Appendix B. Only social behaviors involving another adult macaque were used in this study; behaviors directed at infant or juvenile macaques or human researchers were not included.

This research complied with protocols approved by the Institutional Animal Care and Use Committee of the University of Puerto Rico (protocol #A6850108) and adhered to the legal requirements of the United States of America.

### 2.3. Genetic Data

We obtained animal pedigrees from a long-term database maintained by the Caribbean Primate Research Center (CPRC). From the founding of the population up through 1992, maternal identity was ascertained by behavioral observations, such as nurturing behaviors and lactation. For most macaques born after 1992, both maternal and paternal identity were ascertained genetically through the analysis of 29 microsatellite markers. Previous studies revealed a 97.4% agreement rate between maternity determined by behavior and by genetics [5]. The population pedigree was used to generate a kinship matrix for the animals in the present study via the R package kinship2 [20]. The kinship matrix was then element-wise multiplied by two to create the genetic covariance matrix used to estimate the heritability of behavioral states.

### 2.4. A Model for Social Phenotypes

At the heart of the model is a finite set of latent behavioral states that determines the rates at which different behaviors occur. An animal’s social phenotype is represented in the model as the rate at which that animal experiences the different behavioral states. We describe each component of the model in detail below. In this section, we first present a conceptual description of the model, and the description of the mathematical implementation follows in Section 2.6.

Each ten-minute focal sample is assumed to belong to one and only one behavioral state; that is, during a given focal observation, the focal animal is assumed to be in a particular behavioral state. Each state is defined by a set of parameters that determines the probabilities that each of the discrete behaviors will occur, and, should they occur, how frequently or in what amount they will occur during a focal observation. In the data analyzed here, we used a categorical distribution over various levels of behavioral quantities, but the algorithm can trivially be modified to use different data likelihoods, such as a Poisson distribution for data in count form, a normal distribution for continuous data, and so on. For details on the coding of discrete behaviors in the current study, see Section 2.5 below.

Every animal has a different probability distribution over behavioral states, which describes how often it finds itself in each of those states. This probability is what we call the animal’s social phenotype. An intuitive interpretation of this aspect of the model is that at the beginning of a focal observation, the focal animal flips a *K*-sided die where *K* is the number of states in the model. Whichever side the die lands on determines which behavioral state the animal will be in for the duration of the focal observation. While the *K* behavioral states are common across all animals, each animal has a unique die with different probabilities of landing on any given side compared to any other animal.

Ultimately, we are interested in not just describing social phenotypes and behavioral states, but also understanding how they are associated with other variables of scientific interest, be they morphological, demographic or genetic. Accordingly, our model allows the phenotypes of each animal to be influenced by covariates. We accomplish this by having the probabilities in the animals’ phenotype be determined by a mixed-effects multinomial logistic regression. The regression model incorporates the influence of covariates as fixed effects. Two random effects components are also included in the model. The first describes the intercepts of each of the *K* states, which determine how typical each state is across the population as a whole. The second describes animal-specific error terms for each state, which capture how each animal deviates from the predicted phenotypes based on its covariates and the population average. This mixed effect formulation allows the model to refine the predicted phenotypes for each individual animal by pooling information across the entire population, as represented in the data [21].

In addition to the random effects terms described above, the model can also incorporate a third random effects component based on groupings or relationships among animals. Here we use this term to incorporate genetic effects into the model so that the heritability of phenotypes can be calculated as in the popular “animal model” in behavioral ecology [22].

A major constraint in the model is that during a ten-minute focal sample, the focal animal is confined to only one behavioral state. This constraint is equivalent to assuming that the rate of switching between behavioral states is low enough that the probability of changing states during a consecutive ten-minute period is effectively zero. Though this constraint is not particularly realistic, attempting to infer state shifts within focal samples would dramatically increase the computational complexity of the model. Furthermore, because many individual behaviors are quite sparse—occurring on average at a rate of less than once per focal observation—state transitions within focal samples are likely to be very difficult to detect. We therefore justify this simplifying assumption for the computational efficiency and scalability it permits, though it can be relaxed in the future or with different data.

### 2.5. Data Processing and Likelihood

To construct the input to the model, we calculated the total amount of each behavior present in each observation, and converted those amounts into ordered levels. The exact procedure is described below:
Construct a data matrix with a row for each focal observation, and a column for each behavior in the ethogram.For each focal observation:
(a)For each “event” behavior, count the number of times that behavior occurred during the observation.(b)For each “activity” behavior, calculate the total proportion of the focal observation spent engaged in that behavior.(c)Populate the associated row in the data matrix with these values.For each behavior:
(a)Calculate quintiles, e.g., 20th percentile, 40th percentile, etc., of the values in that behavior’s associated column in the data matrix.(b)Also calculate the 1st and 99th percentile of the behavior to make high and low outliers.(c)Bin using the quantiles calculated above as cutpoints, e.g., values ≤ 1st percentile being 1, > 1st and ≤ 20th percentiles being 2, etc.

This procedure divides each behavior into up to seven levels, with 1 being the smallest amount of a behavior and 7 being the most. However, in practice many behaviors occurred so infrequently that the bottom 50% or more of the data were all zeros. The large number of zeros in most behaviors can be seen in Appendix A. This meant that 19 of 30 behaviors were split into three levels representing 0, 1, and >1 occurrences, and no behavior was divided into more than 6 levels.

A behavioral state consists of a set of categorical distributions, one for each behavior. Each categorical distribution has as many parameters as the behavior has levels that determine the probability of each level of the behavior occurring under that state. This allows each behavioral state to be extremely flexible; a state need not specify a narrow range of values or a specific distributional shape for every behavior. A state may be associated with a very specific amount of one behavior, but also consistent with a wide range of amounts of a different behavior.

The categorical distribution we used as the data likelihood can be easily changed to any distribution with a conjugate prior. For example, one might use the Poisson distribution for event behaviors and the zero-truncated normal distribution for activity behaviors. We chose the categorical distribution here because, as can be seen in Appendix A, many behaviors were zero-inflated with long right tails, which cannot be captured by Poisson and normal distributions.

### 2.6. Mathematical Description of the Model

The content of a focal observation is represented by $y^{\left(i,f\right)}$, a vector of length *B*, where *B* is the number of individual behaviors considered in the model. Element $y_{b}^{\left(i,f\right)}$ is the amount of behavior *b* that occurred in that focal observation. Each focal observation in the data belongs to a single behavioral state. Formally, this means that each observation *f* of animal *i* is associated with a latent variable $z_{i,f}$ which can take on values 1 through *K*, where *K* is the number of behavioral states in the model. This value denotes which of the *K* behavioral states to which the focal observation belongs. Given a value for behavioral state $z_{i,f}$ we can write the data likelihood of the focal observation using the parameters defining the behavioral state:
(1)$$p\left(y^{\left(i,f\right)}|z_{i,f}=k,\theta^{\left(k\right)}\right)=\prod_{b\in 1:B}\theta_{y_{b}^{\left(i,f\right)}}^{\left(k,b\right)}$$

Here $\theta^{\left(k\right)}$ is the set of all parameters associated with state *k*, $\theta^{\left(k,b\right)}$ is the vector of probabilities for different levels of behavior *b* in state *k*, and $\theta_{l}^{\left(k,b\right)}$ is the probability that behavior *b* occurs at level *l* in state *k*. Note that this data likelihood is simply a product of categorical likelihoods, one for each behavior. Because each focal observation is statistically independent conditional on the state assignments, we can write the complete likelihood for animal *i* as simply the product of the likelihoods of the focal observations:
(2)$$p\left(y^{\left(i\right)}|z_{i},\theta \right)=\prod_{f\in 1:n_{i}}p\left(y^{\left(i,f\right)}|z_{i,f},\theta^{\left(z_{i,f}\right)}\right)$$

Here $n_{i}$ is the number of focal observations for animal *i*.

We now turn our attention from the likelihood to the phenotype, which is the prior probability of a focal observation *f* of animal *i* belonging to each of the *k* behavioral states. We can write this prior probability as,
(3)$$\begin{array}{cc} p(z_{i}|\pi_{i}) & =\prod_{k\in 1:K}\pi_{i,k}^{n_{i,k}}\\ n_{i,k} & =\sum_{f\in 1:n_{i}}1\left(z_{i,f}=k\right) \end{array}$$

Here $\pi_{i,k}$ is the prior probability that animal *i* finds itself in state *k* and $n_{i,k}$ is the total number of focal observations of animal *i* belonging to state *k*. The probabilities $\pi_{i,k}$ are themselves determined by via multinomial logistic regression:
(4)$$\begin{array}{c} \pi_{i,k}=\frac{\exp(\eta_{i,k})}{\sum_{k\in 1:K}\exp\left(\eta_{i,k}\right)}\\ \eta_{i,k}=\alpha_{k}+X_{i}^{T}\beta_{k}+u_{i,k}+\epsilon_{i,k} \end{array}$$

Here $\eta_{i}$ is a vector of unbounded propensities for animal *i* to fall into each behavioral state, which are transformed by the softmax function into the probability distribution over states, $\pi_{i}$. $X_{i}$ is a vector of animal-specific covariates, with the state-specific fixed effect regression coefficient vectors $\beta_{k}$. The quantities α, *u*, and ϵ are random effects terms representing baseline state propensities, genetic effects, and individual animal effects, respectively. These random effects are given normal distributions with the variances as free parameters to be estimated:
(5)$$\begin{array}{c} \epsilon_{i,k}∼N\left(0,\sigma_{k}^{2}\right)\\ \alpha_{k}∼N\left(0,\tau \sigma_{k}^{2}\right)\\ u_{\cdot ,k}∼N\left(0,\gamma_{k}\sigma_{k}^{2}A\right) \end{array}$$

The parameters $\sigma_{k}^{2}$ determine how much variability exists across animals in the propensity for state *k* (outside of variability accounted for by the covariates $X_{i}$). The parameter τ controls the extent to which the states themselves vary in their average propensities across animals. That is to say, in a model where the $\sigma_{k}^{2}$ are large and τ is small, no state will consistently have high or low probability across the population, whereas when $\sigma_{k}^{2}$ are small and τ is large, states will have similar probabilities across animals but some states will be consistently high probability and others low. Finally, the covariance matrix *A* is the relatedness matrix of the animals in the study population. The parameters $\gamma_{k}$ determine how much variation in each state’s propensities is accounted for by genetic effects. Further, note that the variance component parameters τ and $\gamma_{k}$ are being scaled by the “global” variances $\sigma_{k}^{2}$. This causes the linear regression in Equation (Equation 4) to be fully conjugated, such that the sampler can be “partially collapse” during inference by integrating out the components of the linear regression [23]. See [24] for an explanation of partially collapsed samples in mixed effects regression.

As we are using a fully Bayesian approach, we must specify priors for the free parameters. We use standard conjugate uninformative or weakly informative priors in all cases:
(6)$$\begin{array}{cc} \theta^{\left(k,b\right)} & ∼\mathrm{Dirichlet}(1)\\ \beta_{k} & ∼N(0,1)\\ \sigma_{k}^{2} & ∼\mathrm{InvGamma}(0.005,0.0005)\\ \tau & ∼\mathrm{InvGamma}(0.005,0.0005)\\ \gamma_{k} & ∼\mathrm{InvGamma}(0.165,0.0165) \end{array}$$

See [25] for a discussion on weakly informative priors, though we do not use their exact priors. Note that the seemingly weakly informative prior on the $\gamma_{k}$ parameters was chosen in order to achieve an uninformative (high variance) prior on heritability, which in this model is approximately $\gamma_{k}/\left(1+\gamma_{k}\right)$. An uninformative prior such as $\gamma_{k}∼\mathrm{InvGamma}\left(0.005,0.0005\right)$ would actually place high prior probability mass on values of $\gamma_{k}/\left(1+\gamma_{k}\right)$ near 1. The chosen prior is roughly symmetric on $\gamma_{k}/\left(1+\gamma_{k}\right)$ and places much of the probability mass near both 0 and 1.

### 2.7. Model Fitting

We fit the model using a custom Gibbs sampler implemented in the Julia technical computing language v0.5.0 [26]. As standard logistic regression representation is non-conjugate and therefore cannot be sampled from using Gibbs, we use the fully conjugate latent variable formulation of logistic regression described in [27] and previously applied to topic models in [9]. In the results reported below we ran two chains of 100,000 samples each, which were thinned to 1000 samples, with the first 100 of those discarded as burn-in.

Our model involves unsupervised classification with several hundred parameters, which means the posterior is likely to be highly multimodal. Gibbs sampling with naively chosen random starting points can often get stuck around suboptimal and idiosyncratic local maxima, which leads to both poor inference and results that are unpredictable and unreproducible between runs. Tests with random starting points indicated that even in simple synthetic data sets, where observations fall into distant, non-overlapping clusters, Gibbs sampling alone very frequently yielded incorrect clustering whenever more than three or four behavioral states were used. In many applications of topic modeling, reproducibility and interpretability of model outputs are of secondary concern to pure predictive performance, but for scientific inference they are paramount.

In order to improve the quality and reliability of inference we first fit a simpler version of the model which we then used to generate starting points for the full model. Specifically, we fit a “flattened” version of the full model which is equivalent to assuming that all observations came from the same animal, thus discarding all information about differences between animals and population level covariates. This model is in effect a naive Bayes classifier, where each possible classification is analogous to a behavioral state. This flat model was fit by maximum likelihood (ML) using the Stan software package v2.14.0 [28]. The initial points for this optimization step were generated by first taking the ML solution for a model with a single behavioral state, then adding independent Gaussian noise to each parameter to generate starting points for each behavioral state. Multiple fits with independent initializations were generated and the fit with the highest posterior density was used to initialize the Gibbs sampler. Specifically, for each focal observation, an assignment to a behavioral state was randomly drawn from the posterior distribution of behavioral state memberships under the ML fit, and this mapping from observations to behavioral state was used as the starting point for the Gibbs sampler. We found that in practice this procedure very reliably recovered the true parameters when applied to simulated data and provided consistent results using real data.

In order to choose an appropriate number of behavioral states, *K*, we calculated the widely-applicable information criterion (WAIC) [29] for models (without a genetic component) with 5, 10, 15, and 20 states, and used the number of states associated with the lowest WAIC. See Appendix A for the WAIC results.

### 2.8. Repeatability Analysis

Seventy-seven animals from social group F were observed for two consecutive years (2012 and 2013), and we used these animals to assess the stability of the phenotypes discovered by our model. To accomplish this objective, we ran a separate model in which all animals from the previous analysis were included, but animals with dual observation years were permitted to have independent phenotypes for each year. We then examined the correlation between the posterior means of the phenotypes in 2012 and 2013. No heritability component was included in this model. Note that this version of the model was not used for any analysis other than heritability, as the artificial inflation of the number of animals might lead to unwarranted confidence in population-level inferences.

### 2.9. Simulations

To verify that the behavioral state model could correctly recover both state contents and individual phenotypes, we tested the model on synthetic data in series of five simulations using 5, 10, 20, 40, and 80 states. For each simulation, we simulate 200 individuals with 100 focal observations each, using an ethogram consisting of 30 behaviors. Each behavior was represented as a categorical distribution with 3 levels. States were generated by sampling from the following prior:
(7)$$\begin{array}{cc} \lambda_{l}^{\left(k,b\right)} & ∼N(0,1.0)\\ \theta_{l}^{\left(k,b\right)} & =\frac{\exp(\lambda_{l}^{\left(k,b\right)})}{\sum_{l^{\prime }\in 1:3}\exp\left(\lambda_{l^{\prime }}^{\left(k,b\right)}\right)} \end{array}$$

Similarly, individual phenotypes were generated by sampling from the following distribution:
(8)$$\begin{array}{cc} \eta_{i,k} & ∼N(\alpha_{k},0.0625)\\ \alpha_{k} & ∼N(0,0.25) \end{array}$$

These values are converted into individual probability distributions over states. Each individual’s probabilities are then used to sample a state membership for each of that individual’s 100 focal observations.

Because states have no intrinsic ordering, some method is required for associating states in the simulated data with a matching state in the model output before it is possible to determine whether a state or phenotype has been accurately estimated. To accomplish this, we used a greedy matching algorithm with the folowing steps:Pick an output state $k^{\prime }$ and calculate the posterior mean for each of its parameters, ${\hat{\theta }}^{\left(k^{\prime }\right)}$.For each simulated state *k* that is not already matched with an output state, calculate the correlation between $\theta^{\left(k\right)}$ and ${\hat{\theta }}^{\left(k^{\prime }\right)}$.Pick the simulated state with the highest correlation as the match for the output state $k^{\prime }$.Repeat for each $k^{\prime }$.

After the matching procedure, we assessed how well the model recovered individual phenotypes by calculating for each individual the correlation between that individual’s simulated probability of being in each state and the probabilities of being in the matching states of under the fitted model. We also assessed our ability to recover the true number of behavioral states in the 5-state simulation by comparing WAIC scores of the model with 5 states to models with 3, 4, 6, and 7 states fitted to the same data.

The model was fit to the simulated data using the same fitting procedure as used with the Cayo data. However, no regression coefficients or heritabilities were calculated.

### 2.10. Comparisons with Factor Analysis

We fit factor analysis models to the Cayo Santiago data set to compare with the behavioral state model presented here. As factor analysis and related models have no explicit hierarchical structure, they cannot separate observation-level relationships among behaviors from organism-level relationships; they can either examine how behaviors co-occur within observations, or how average rates of behaviors can co-occur between macaques, but not both simultaneously. Therefore we fit two models. The factor model 1 used the focal observations themselves as independent data points. For factor model 2, we calculated average rates of each behavior for each macaque and used macaques as independent data points. To set the number of factors used, we used 5-fold cross-validation to choose amongst 5, 10, 15, and 20 factors. For both versions 1 and 2 of the model, models with 10 factors or more performed similarly, so we used 10 factors for ease of comparison with the state model. We defined the phenotypes infered by the factor models as the factor scores of the individual animals. In the case of factor model 1, scores are associated with individual focal observations rather than individual animals, so the phenotypes were the average of the scores across each macaque’s focal observations.

To estimate heritability of phenotypes under both models, we fit an animal model to the inferred phenotypes. These models used the same covariates as the state model.

To calculate conditional means for rates of behaviors under factor model 1, we sampled focal observations from a multivariate normal distribution with the means and covariance matrix fit by the factor model. Since the true data is discrete rather than continuous, for each of the sampled observations we rounded the behavior amounts to the nearest whole number occurring in the true data.

All factor analyses were carried out using the (factanal) function in R 3.4.0 using “regression” scores. [30]. Animal models were written and fit using the Stan software package [28].

### 2.11. Assessing Genetic and Covariate Influences on Social Phenotypes

In multinomial logistic regression, regression coefficients associated with a state are only interpretable relative to a baseline state [31]. This is due to the fact that a probability distribution over *K* states has only K-1 free variables, as the probability of a single state is determined completely by that of the remaining states. If a baseline state is not specified, parameters will be unidentifiable with respect to the data and will be constrained only by the prior. Regression coefficients and genetic effects, with the coefficients and genetic effects of the baseline state subtracted, therefore reflect an influence on the relative probabilities of a each state occurring versus the baseline state.

A related issue is that the state probabilities, here the phenotypes of interest, are a nonlinear function of the underlying model parameters and covariates. The impact on the phenotypes of any given component of the model depends on the value of every other covariate and parameter in the model. Due to this nonlinearity, and because values depend on the baseline selected, parameter estimates themselves can be very difficult to interpret. However, an advantage of our Bayesian approach is that we can easily derive estimates and central credible intervals (CIs) for the influence of specific components of the model on the phenotypes of interest. We accomplished this by generating predicted phenotypes from posterior samples under different assumptions to assess the contributions of different components of the model. To assess the impact of a specific covariate on phenotypes, we generated predicted state probabilities at varying levels of that covariate while holding every other covariate fixed at its population average value (or, in the case of discrete covariates, the modal level). This process was repeated for every posterior sample of the model parameters, yielding a full posterior distribution of predicted phenotypes.

To assess which, if any, behavioral states were strongly influenced by genetic effects, we calculated a pseudo-$h^{2}$ metric to describe the amount of variance in state probabilities that was explained by genetic effects and covariates combined relative to the variance captured by covariates alone. Formally, for state *k* the total variance in state probabilities is $\mathrm{var}(\pi_{\cdot ,k})$, where $\pi_{\cdot ,k}$ is the vector of state probabilities as defined above (Equation (Equation 4)). We can also define partial variances based on state probabilities estimated when certain components of the model are omitted:
(9)$$\begin{array}{c} {\hat{\pi }}_{i,k}^{\left(B\right)}=\frac{\exp({\hat{\eta }}_{i,k}^{\left(B\right)})}{\sum_{k\in 1:K}\exp\left({\hat{\eta }}_{i,k}^{\left(B\right)}\right)}\\ {\hat{\eta }}_{i,k}^{\left(B\right)}=\alpha_{k}+X_{i}^{T}\beta_{k} \end{array}$$
(10)$$\begin{array}{c} {\hat{\pi }}_{i,k}^{\left(u\right)}=\frac{\exp({\hat{\eta }}_{i,k}^{\left(u\right)})}{\sum_{k^{\prime }\in 1:K}\exp\left({\hat{\eta }}_{i,k^{\prime }}^{\left(u\right)}\right)}\\ {\hat{\eta }}_{i,k}^{\left(u\right)}=\alpha_{k}+X_{i}^{T}\beta_{k}+u_{i,k} \end{array}$$

We then define pseudo-$h^{2}$ for behavioral state *k* as,
(11)$$1-\frac{\mathrm{var}(\pi_{\cdot ,k}-{\hat{\pi }}_{\cdot ,k}^{\left(u\right)})}{\mathrm{var}(\pi_{\cdot ,k}-{\hat{\pi }}_{\cdot ,k}^{\left(B\right)})}$$

This is the proportion of variance explained by adding genetic effects back into a model, out of the residual variance left over by a model with covariates alone. This is both closely related to a partial $R^{2}$, and to the traditional $h^{2}$ measure used in animal models, where the latter is commonly defined as the proportion of variance explained by genetic effects out of the residual variance after the effects of covariates are removed [32]. As with covariates, we calculated pseudo-$h^{2}$ for every posterior sample. Finally, we note that unlike standard $h^{2}$ and $R^{2}$, pseudo-$h^{2}$ can be negative, indicating that including genetic components reduces predictive accuracy.

## 3. Results

### 3.1. Simulation Tests

We first verified that our model could accurately recover both the contents of behavioral states and phenotypes of individual animals by fitting the model to simulated data sets and comparing the model outputs to the ground truth. Figure 1 shows that for data sets of similar size as the Cayo Santiago data, the model accurately recovered all behavioral states so long as the number of states did not exceed 20. In each simulated data set with up to 20 states, every simulated state had a unique state in the fit model for which their state parameters $\theta^{\left(k\right)}$ had a correlation coefficient above 0.95. When the number of states exceeded 20, the majority of simulated states had a close match in the model’s estimated states, but several simulated states had no match with correlations above 0.5. This suggests that the model was unable to find some behavioral states.

Similarly, the ability of the model to reliably recover phenotypes declined as the number of states increased. With 5 and 10 states, the correlation between fitted and simulated phenotypes was above 0.75 for most individuals and above 0.5 for nearly all of them, while in the data sets with 40 and 80 states most individuals had correlations below 0.5. This is not surprising, as larger numbers of states means more parameters must be estimated for each individual using the same number of observations.

We also found that for the data set with five states were were able to identify the model with five behavioral states as the optimal model using WAIC goodness-of-fit metric. See Appendix A for the WAIC results.

### 3.2. Phenotype Distributions and Behavioral State Content

We fit the model allowing ten behavioral states (referred to hereon as S1–S10). Figure 2 shows the distribution of phenotypes across the population, in the form of the rate at which each animal exhibited each behavioral state. We ordered the states in terms of how frequently each state occurred in the population, with S1 having the highest average rate of occurrence and S10 having the lowest. S1–S3 showed both high overall rates, with animals spending on average 55% of their focal observations in these states, as well as high variability in their rates across animals, with substantial numbers of animals spending more than 30% of their focal observations in a single one of these three states. S4–S9, on the other hand, all had median rates falling in a narrow range between 7% and 5.6%, with only one animal displaying a rate greater than 20% in any one of those states. Finally, S10 has a median rate of only 1.4%.

Next we examined the contents of these behavioral states. Figure 3 visualizes the typical behaviors of each state in terms of their relative frequency, with very rare behaviors omitted for legibility. Such “at a glance” summaries distinguish particular features of each behavioral state. Immediately apparent is the distinction between the more frequent S1–S3 and the infrequent S4–S9. S1–S3 paint a rather prosaic portrait of macaque life, consisting largely of self-directed behaviors, such as scratching oneself, eating, and walking. Indeed, S2 entailed doing little other than resting. The social interactions that did occur in these three states were primarily agonistic. In particular, both overt non-contact aggression and less overt agonistic actions, such as fear grimaces and avoidances, were quite common in S1. The middle infrequent S4–S9 on the other hand, displayed mixtures in varying proportions of agonistic and affiliative actions. Incidental affiliative behaviors such as approaching other animals occurred commonly throughout all of these infrequent states, while grooming, a more significant sign of affiliation, appeared concentrated in S4, S5, and S8.

### 3.3. Repeatability

As our goal is to identify phenotypes with underlying biological bases, it is important to show that phenotypes produced from the model are consistent within animals and relatively stable across time. To that end we compared estimated phenotypes from animals in group F from years 2012 and 2013. Phenotypes in 2012 were strongly correlated with phenotypes in 2013 for all behavioral states, with the lowest correlation coefficient being 0.64 for S1, and the largest being 0.88 for S8 (*p* < 0.001 for both).

### 3.4. Group, Rank, and Sex Effects on Social Phenotypes

While describing the content of behavioral states is useful, scientists are often more interested in identifying sources of behavioral variability. In rhesus macaques and other NHPs, it is important to understand how much variability in social phenotypes across animals can be predicted on the basis of demographic covariates, and how much is idiosyncratic to each individual. In the present model we included age, sex, dominance rank, social group, and age-by-sex and rank-by-sex as population-level predictors of phenotypes. Dominance rank was represented on an ordered categorical scale: low-ranking animals outranked less than 50% of their social group, medium-ranking animals outranked between 50% and 80%; and high-ranking animals outranked greater than 80%. Ranks were available annually, so for animals with multiple years of observations that changed ranks, their average rank was used. Figure 4 displays the regression coefficients for each covariate’s influence on the probability of being in each behavioral state. Here we choose S2 as baseline as it provides a neutral default state with little in the way of positive or negative social interactions.

The posterior distributions of the regression coefficients indicate that social group membership and sex had large influences on social phenotypes. Animals in groups R and HH had, for instance, lower relative rates of being in S1 versus S2 than animals in the largest social group, F, with regression coefficients of −0.52 ([−0.78, −0.27] 95% CI) and −0.88 ([−1.20,−0.58] 95% CI), respectively. As S1 had much higher rates of agonistic behavior than the unsocial S2 and typically involved very little affiliative behavior, this suggests that animals in group F were more likely to become involved in agonistic interactions. Sex and linear dominance rank were the most common predictors of relative behavioral state occurrence rates, with four and five of the nine non-baseline states respectively showing significant effects. These states overlap, with S1, S3, and S6 being more frequent in male macaques and in lower ranking monkeys, which we note were states with higher rates of agonistic behaviors and relatively lower rates of affiliative behaviors.

Though examining the posterior of regression coefficients predicting relative odds of states can be useful, it is often more intuitive to directly examine the influence of covariates on the absolute rates at which an animal experiences behavioral states. We used this method to examine how sex, group membership, and dominance rank influenced phenotype. Figure 5 displays the influence of social group membership on the social phenotype of male and female animals of median age and rank. This representation of the model demonstrates that males experienced S1 and S3 more frequently than females, at the expense of slightly lower rates of all of the less frequent, more social S4–S10. S1 entailed giving and receiving aggression while moving and feeding, while S3 consisted of self-directed grooming and scratching with some amount of agonistic actions that may have served to defuse aggression. Both featured low rates of affiliative social behavior. In contrast, the states males experienced less often contained higher relative rates of affiliative behaviors and peaceful sharing of space with conspecifics. This implies that males spent more of their time avoiding confrontations and competing for resources and less time experiencing social support.

It is worth noting that males experienced S6 at lower rates than females did, despite the fact that the S6 regression coefficient for ‘male’ has a positive posterior mean and little posterior mass below zero (0.13 posterior mean, [−0.03,0.45] 95% CI). This illustrates the value of examining absolute state probabilities rather than regression coefficients themselves. In other words, because males spent so many focal observations in S1 and S3, the positive coefficient for males in S6 reduced the extent to which S1 and S3 crowded out S6 for males, but did not reverse the gap between sexes.

Finally, we note that S10, the least frequent of states overall, showed a dramatic difference between groups. Monkeys in group R experienced this state far more often than members of either F or HH. As before, this effect was not apparent from the raw regression coefficients, as the coefficients for HH and R membership were similar in absolute value, with posterior means of −0.91 ([−1.80,−0.14] 95% CI) and 1.45 ([0.96, 1.95] 95% CI) respectively. S10 was characterized by very high rates of feeding while maintaining close proximity to large numbers of other animals. Focal animals in S10 also engaged in high rates of agonistic behaviors such as threats and non-contact aggression, suggesting this state captured episodes of food competition within or between groups.

Figure 6 shows the influence of dominance rank on the phenotype of an animal of median age in group F. Here we observed an overall pattern across behavioral states that was similar to the effect of gender, in that S1 and S3 were overall more frequent among low ranking animals than high, while S4–S10 were more frequent in high ranking animals. We also observed strong rank-by-sex interactions in S3 and S9, whereby higher rank had little effect on the rate of being in S3 for females but greatly decreased that rate for males, while the opposite was true for S9.

### 3.5. Genetic Components of Social Phenotypes

Finally, we sought to clarify which aspects of social phenotypes were strongly influenced by genetic factors. To quantify how much variability in the probability of being in each behavioral state could be accounted for by genetic factors, beyond the variability explained by demographic covariates alone, we calculated pseudo-$h^{2}$ as described above for each behavioral state and displayed the posterior distributions in Figure 7. For most states, we found very wide 95% credible intervals for the variability explained with the lower bounds of eight of the ten states falling below 10%. For these states, we could not effectively rule out negligibly small contributions of genetics in determining behavioral state probabilities. However, S5 and S6 showed reliably high pseudo-$h^{2}$ of 0.69 ([0.43, 0.80] 95% CI) and 0.6 ([0.24, 0.76] 95% CI), respectively.

S5 was associated with high levels of affiliative behavior, involving high rates of friendly vocalizations, grooming, and passive physical contact, as well as being in close proximity to a high number of conspecifics for long periods of time (Proximity Group Size and Social Proximity, respectively, see Appendix B for definitions). S5 also showed low rates of most agonistic interactions. Conversely, S6 was characterized by a more diverse selection of behaviors, entailing high rates of less confrontational forms of agonistic behavior (e.g., giving and receiving fear grimaces and displacements) and more actively confrontational behavior (e.g., giving threats and noncontact aggression, as well as submissive gestures). Nonetheless, animals in S6 were not deterred from affiliative approaches.

### 3.6. Comparison with Factor Analyis

One advantage of the state model over factor analysis and related methods is that it can capture a wider range of relationships among behaviors. Figure 8a gives an example of this using the behaviors Travel, Feed, and Noncontact Aggression (received). The relationship between traveling and feeding is nonlinear, with the rate of traveling increasing with the rate of feeding for low levels of feeding, but sharply decreasing at very high rates of feeding. Moreover, this relationship is moderated by noncontact aggression received. During focal observations when the animal is not feeding, animals travel much more when receiving aggression, but when feeding occurs received aggression has little impact on time spent traveling. The state model captures both the nonlinear relationship between travel and feeding as well as the interaction with noncontact aggression received, though it does notably underestimate how quickly time spent traveling decreases at high levels of feeding. Factor model 1, however, can only represent linear relationships [33] and so captures only the positive correlation between traveling and feeding at low levels of feeding. We did not assess factor model 2 in this way; because factor model 2 is a model of average behavior rates by individuals rather than a model of focal observations themselves, we saw no clear method for generating predictions for these observation-level relationships.

Finally, we also compared the state model to the factor models in the repeatability and heritability of the phenotypes generated by the model. As shown in Figure 8b, the phenotypes under the state model showed higher repeatability than those of either factor model. As shown in Figure 8c, the factor model yielded no phenotypes with heritability estimated at above 0.5, while the state model and factor model 1 phenotypes showed broadly similar distributions of heritabilities. However, given the high uncertainties associated with heritability estimates under all models, any conclusions about differences among models must be tentative at best.

The loadings associated with factor models 1 and 2 are shown in Appendix A, respectively.

## 4. Discussion

The influence of typical biological and environmental factors on social behavior is likely to be many-to-many, with any given factor impacting many different behaviors together rather than any specific behavior in isolation [34,35,36,37]. Dealing with high-dimensional, structured behavioral data will therefore be important for understanding the biological bases of behavior. The field of primatology has decades of experience collecting detailed, high-dimensional data on social behavior in species with complex, human-like social behaviors, and this data may prove invaluable for linking fundamental biology to complex social behavior. However, extracting useful structure from natural, high-dimensional and relatively noisy behavioral data sets remains a major challenge. In this paper we presented a model for identifying patterns of social behavior and relating them to environmental and genetic factors, as inspired by machine learning methods for uncovering underlying topics in corpora of documents.

The behavioral states our model discovered and their relationships with demographic covariates were consistent with our knowledge of rhesus macaque behavior generally. Macaques on average spent about 50% of their time in S1–S3, which involved little social interaction beyond receiving non-contact aggression and other agonistic encounters in S1. This pattern may reflect conflict over food or space given that state’s high levels of feeding and traveling. The other 50% of the observed time was evenly distributed across the remaining states, S4–S10, which consisted of idiosyncratic combinations of agonistic and affiliative behaviors, reflecting the long-known importance of sociality in rhesus macaque life [15,19]. Among both males and females, higher dominance rank was associated with a shift away from the frequent S1 and S3, which involved mostly self-directed behavior amidst some agonistic interactions, and towards S4–S10 which had higher rates of many different social interactions overall as well as high rates of affiliative interactions. This is consistent with the many previous findings that high ranking macaques, and other primates as well, receive higher rates of affiliative behaviors from others in their social group, while low ranking individuals are more likely to experience social isolation [13,15,16,38,39,40]. The effect of sex on predicted phenotypes was similar to the effect of dominance rank, with males, like low-ranking animals, having higher probabilities of assignment to S1 and S3 at the expense of time spent in S4–S9 (but not S10, notably). This may reflect the inheritance of female rank in rhesus societies, with females remaining in their natal groups and acquiring a rank just below their mother. By contrast, males disperse into new social groups at adulthood and must earn their ranks there. This results in dominance hierarchies generally being more stable, and therefore social relationships more peaceful, among females than males [41,42].

Our model improves upon the commonly used factor analysis and PCA methods in that it captures information regarding the way individual behaviors co-occur. First, factor analysis and related methods implicitly assumes that data is normally distributed [33]. This assumption is strongly violated by observations of natural behaviors, which are typically represented as counts. Furthermore, many behaviors of interest are relatively infrequent (see Section 2.6 on data processing and likelihood), leading to highly skewed, zero-inflated distributions for individual behaviors. Our model uses instead a flexible discrete distribution to avoid mismatches between the assumed data distribution and the data itself (though any distribution with conjugate priors can be substituted with minimal effort). Second, and more because of the multivariate normal assumption, PCA and factor models can only represent relationships between two behaviors as correlations. The behavioral state model, being a type of mixture model, is more flexible and capable of capturing nonlinear relationships and interaction effects [43]. This means PCA cannot capture nonlinear relationships between pairs of behaviors, or relationships which are modulated by a third behavior. The seemingly quadratic relationship between time spent feeding and traveling in Cayo Santiago macaques, which itself depends on whether aggression is received, is an example of the kind of complex relationship among behaviors that that the behavioral state model is able to represent.

We are not the first to apply topic model-like methods to primate behavioral data. The popular program STRUCTURE, which was in fact developed prior to topic models, represents the genotypes of organisms as distributions over distinct genetic populations [44], just as topic models treat documents as distributions over topics and just as our current model describes the phenotype of an animal as a distribution over behavioral states. This model has been applied to distinguish Indian and Chinese rhesus macaque genotypes [45]. The current model is, to the best of our knowledge, the first attempt to apply topic models to natural animal behavior.

In this paper we used a pedigree to estimate additive genetic influences on behavioral phenotypes, as per classical heritability analyses. An alternative approach is to include individual genetic variants as predictors in the model as in genome-wide association analyses (GWAS). Our model’s regression layer makes GWAS very straightforward to implement, as variant data such as minor allele count can be included directly in the model as a per-animal covariate. However, while this approach would yield much more precise information on how different genetic variants influence phenotypes, it may be difficult to implement for two reasons. First, the traditional GWAS methodology involves fitting a separate model for many common genetic variants [46]. While this is tractable when the model being fit is a standard linear regression, fitting a complex hierarchical model thousands of times is computationally infeasible. Second, the sample sizes available in observational data sets of wild or free-ranging animals reach the hundreds or low thousands at best, which are orders of magnitude too small to achieve adequate power to detect the generally small effects of common genetic variants [47,48,49]. However, inference on the effects of rare or de novo mutations, which can have larger effect sizes, may be more feasible [37,50,51]. Gene sets [52] or genetic risk scores, which aggregate known genetic effects on biological pathways, provide alternative approaches.

Another possibility is to use realized relatedness based on measured genetic similarity to estimate genetic influences on phenotypes. This method assumes that many genetic variants have some small effect and aggregates information across all of them rather than attempting to identify specific variants with large effects [34,53]. This assumption is highly appropriate for complex behavioral phenotypes such as social behaviors, which are known to be polygenic in humans [35,36]. While aggregating across many variants means individual variants cannot be singled out as important for a given phenotype, recent methodological advances suggest that analyses based on realized relatedness can be used to decompose overall genetic influences into the influences of specific genomic regions of interest, thus retaining some specificity and allowing insight into the roles of different genetic pathways [54,55,56]. Several groups have begun investigating the use of realized relatedness to examine genetic contributions to phenotypes in natural populations [57,58], though to the best of our knowledge no groups have yet examined the contributions of different genomic regions or pathways. This may present a path forward for understanding the genetics of natural behavior, when combined with models of behavior such as that developed here that effectively and efficiently aggregate information both across behaviors and animals.

## 5. Conclusions

As neuroscientists and biologists investigate the biological determinants of natural social behavior and attempt to translate laboratory findings into more realistic, unconstrained environments, they face the perennial challenge of quantifying natural social behavior. Because natural social behavior is high-dimensional, highly variable, and yet highly structured, straightforward measurement of its influences is difficult. We have worked to address this issue by developing a model for identifying patterns of social behavior and relating them to environmental and genetic factors. Based on recent advances in machine learning for identifying latent structure in sets of documents, this method captures a wider range of relationships among behaviors than is possible using popular methods such as factor analysis or PCA, and explicitly disentangles variability in an individual’s social behavior from variability across the population. We hope that this model will aid researchers in quantifying social behaivior in a way that is rigorous and consistent, while also capturing the richness and complexity of social behavior.

## Figures and Tables

**Figure 1 brainsci-07-00091-f001:**
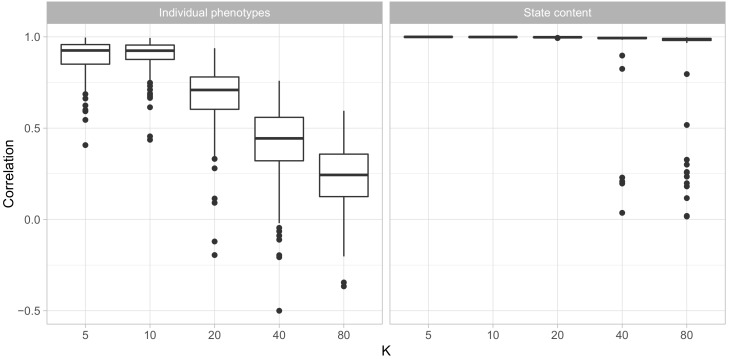
Correlations between simulated and fit phenotypes (**left**) and state contents (**right**). Both panels show boxplots, though for the state contents the values are concentrated enough that the hinges of the plots are not distinguishable. For phenotypes, each data point is an individual, while for states, each data point is a state.

**Figure 2 brainsci-07-00091-f002:**
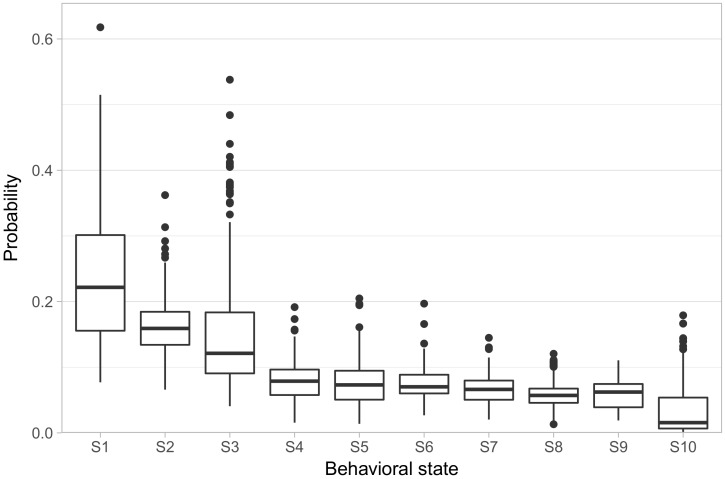
The distribution of phenotypes of the studied population. The box-and-whisker plots display the distribution of posterior mean probabilities of being in each state across all studied individuals. States are ordered by descending mean probability across the population. “Hinges” of the boxes represent 25%, 50%, and 75% quartiles.

**Figure 3 brainsci-07-00091-f003:**
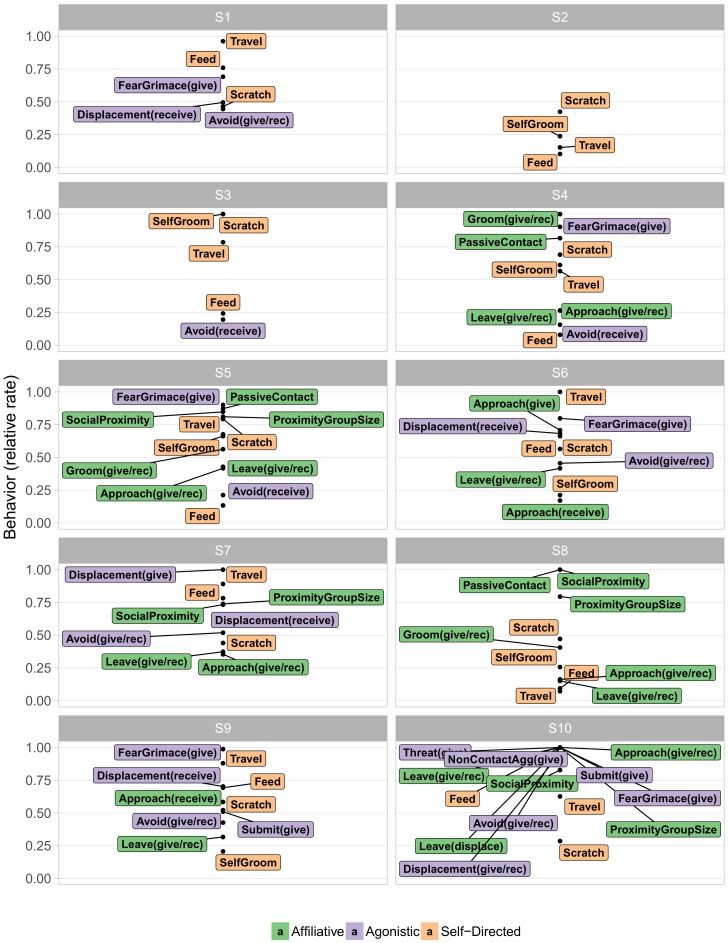
The content of behavioral states fit by the model. Relative rates for each behavior are calculated by dividing the posterior mean rates across states by the mean rate of the highest state, such that 1 represents the highest mean rate across states. For visual clarity, behaviors with relative rates below 0.05 are omitted, and behaviors for which the difference between the “give” and “receive” variants is less than 0.33 are concatenated into a single label (give/rec).

**Figure 4 brainsci-07-00091-f004:**
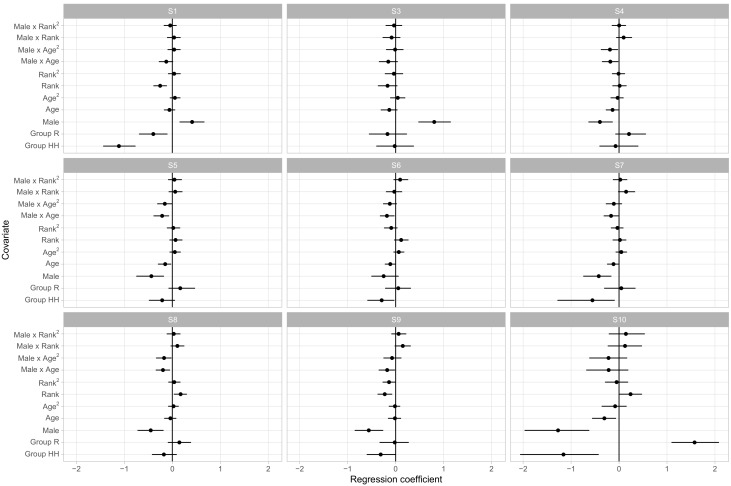
Regression coefficients with S2 as a baseline (see Methods). Points represent posterior means and lines, 95% central credible intervals.

**Figure 5 brainsci-07-00091-f005:**
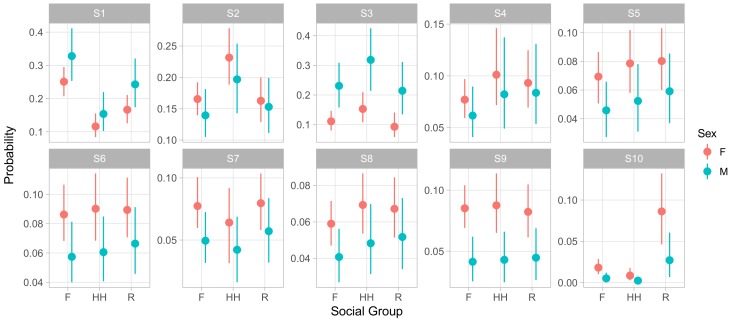
Expected probabilities of being in a state across different social groups for animals of average age and rank. Points represent posterior means and error bars, 95% central credible intervals of the expected probability.

**Figure 6 brainsci-07-00091-f006:**
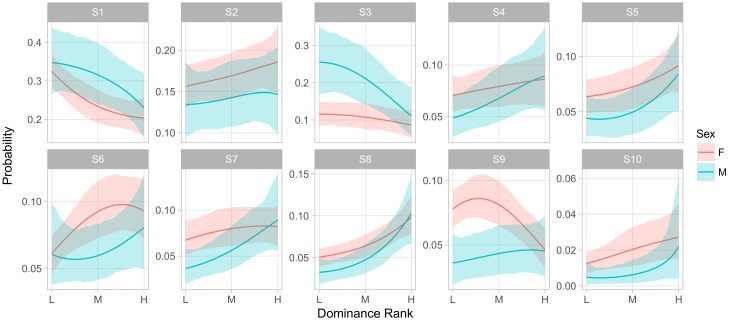
Relationship between dominance rank and the expected probability of being in a state for animals of average age in group F. Curves represent posterior means and shaded regions, 95% central credible intervals of the value of the curve at the corresponding rank.

**Figure 7 brainsci-07-00091-f007:**
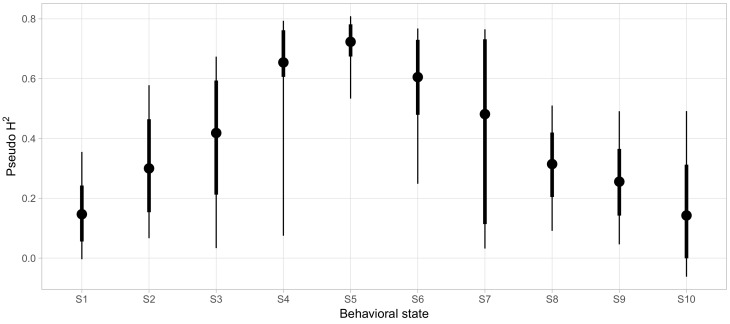
Genetic influences on the probability of being in a state in the studied population. Points represent posterior means, thick error bars, 66% central credible interval (roughly equivalent to 1 standard error), and thin error bars, 95% central credible intervals. See Methods for definition of the pseudo-h2 measure.

**Figure 8 brainsci-07-00091-f008:**
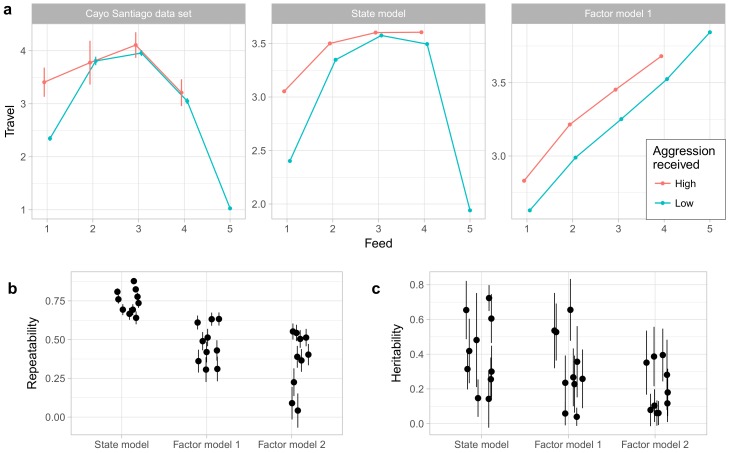
Comparisons between the state model and factor analysis models. (**a**) The leftmost panel shows observed means and standard errors of rates of Travel at varying levels of Feed and Noncontact Aggression (received); The right two panel shows the expected levels of Traveling under the state model and factor model 1; (**b**) Repeatability, as measured by the correlation between 2012 and 2013 phenotypes, for the three models; (**c**) Heritability estimates of phenotypes from the three models. Error bars in all panels represent one standard error.

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
