# Peer review of "Using Machine Learning to Discover Latent Social Phenotypes in Free-Ranging Macaques"

_brainsci, 2017, doi:10.3390/brainsci7070091_

Round 1

Reviewer 1 Report

This paper uses a novel machine learning model to behavioral sets from a large pool of observations of free ranging rhesus macaques.  The behavioral states are subsequently compared to population factors, including rank, sex, group and genetics. The independent sorting identifies 10 behavioral groups that account for a large percent of total behavioral states that are consistent with rhesus ethology.  The model eloquently handles the large multidimensional data sets that arise from longitudinal observational animal studies and presents an improvement on the current PCA based analytical techniques due to its accommodation of non-normally distributed behaviors. There are a couple features that could be added to this paper to make it more accessible to the broader audience of primatologists.

1)    This paper would benefit from a diagrammatic presentation of the data structure to emphasize the pre-processing and the format of the data which goes into the subsequent analyses.

2)    The main strength of this model relative to the current techniques is its handling of non-normal data sets.  It would strengthen the article to have a more direct comparison of how PCA would have differently sorted the behavioral sets. This would be particularly relevant to the a priori interest to affiliative and agonist behaviors.

3)    As this is a longitudinal study, time of day and time of year play large roles in the generation of the behavior.    The model would benefit from accounting for these factors to validate that the behavioral sets are not simply capturing time features (or if they are, acknowledging such).

4)    The assumption of a single behavioral state for each observation is with limitations.  The validity of this assumption could be tested by running the model on a smaller time window within each observation (ie do the same behavioral sets emerge if only the first 5 minutes of each observation are used).

Overall, this paper presents an interesting new computational approach to analyzing large data sets and is very well written. 

Author Response

We thank the reviewer for their helpful suggestions and respond point-by-point below.

1)    This paper would benefit from a diagrammatic presentation of the data structure to emphasize the pre-processing and the format of the data which goes into the subsequent analyses.

After attempting a few such diagrams, we felt that this would be clearer and more space-efficient to present the preprocessing using a list of ‘pseudo-code’ steps. We hope the reviewer agrees!

2)    The main strength of this model relative to the current techniques is its handling of non-normal data sets.  It would strengthen the article to have a more direct comparison of how PCA would have differently sorted the behavioral sets. This would be particularly relevant to the a priori interest to affiliative and agonist behaviors.

This is a very good point that was shared by the other reviewer. We have added new results that we hope clarify how our model differs from PCA and related methods in a more concrete way.

3)    As this is a longitudinal study, time of day and time of year play large roles in the generation of the behavior. The model would benefit from accounting for these factors to validate that the behavioral sets are not simply capturing time features (or if they are, acknowledging such).

Time of day and year are quite certainly predictive of which behavioral state an animal ends up in. Though our estimates of behavioral state content are not biased by the omission of these data, including them as covariates in the model would both provide useful and interesting results and possibly allow us to sharpen our mapping between behavioral states and focal observations. Unfortunately, the model as is currently formulated does not allow within-animal variation in phenotypes. That is to say, we can either estimate phenotypes that vary across animals, or across time of day, but not across both at once! Conceptually and mathematically tweaking the model to allow this is quite straightforward, but due to some unfortunate choices made by the lead author in structuring his code, implementing those changes is non-trivial and we were unable to do so in a timely manner.

We also note that our estimates of individual phenotypes and their relationships with other covariates should not be biased by the omission of time-of-day or season, as the focal sampling schedule was carefully designed to counterbalance observations across these dimensions.

4)    The assumption of a single behavioral state for each observation is with limitations.  The validity of this assumption could be tested by running the model on a smaller time window within each observation (ie do the same behavioral sets emerge if only the first 5 minutes of each observation are used).

This is a good point; to test the reliability of behavioral states within a focal observation we ran a limited version of the model in which the first and second half of each observation was assigned an independent state. In this analysis the most probable state assignments for the first and second half were the same around 40% of the time. While this is vastly higher than the 10% agreement expected by chance with 10 states, it is also not spectacularly high! For the reasons described in the manuscript we still consider this assumption appropriate given the nature of the data and intended application, but ways of relaxing this assumption while maintaining stability and scalability of the model are very much of interest for future work.

Overall, this paper presents an interesting new computational approach to analyzing large data sets and is very well written.

We are pleased that the reviewer found it so!

Reviewer 2 Report

This is a very interesting manuscript describing the use of machine learning, more specifically a topic model, for data reduction and identification of latent behavioral traits in free-ranging macaques. The rationale behind the study is solid, the data set is well suited for this type of investigation and the statistical model is novel and well described. However, I have two main concerns related to questions of the statistical performance of the introduced model and how it compares to more traditional ways of data reduction used in psychology for example.

First, developing a new statistical method, or applying an established method to a new problem, using real world data can to some extent be compared to searching in the dark; it is difficult to assess if the new model captures the “real” underlying data structure since in empirical data true relationships are usually unknown. This is the reason why development of new statistical methods often includes some sort of statistical simulation, allowing data with pre-defined structures to be generated and used to evaluate the performance of the investigated statistical method. Regarding the method in the manuscript under review, I think it would for example be of interest to know how well states are identified in different scenarios like varying levels of residual error and when the true relationship between states are independent vs. non-independent.

Second, although theoretical reasons for why the topic model could outperform more traditional methods for data reduction are mentioned in the manuscript, I think it is essential when proposing a new (and for many readers a more complicated) method that it is clearly shown how this new technique compares to established ways of addressing the same problem. Even though I agree with the authors comments about the limits of using PCA for data reduction in data like this, based on my experience with similar problems, structural equation models (including factor analytic methods) would be a better comparison. The authors mention that researchers using PCA summarize data per individuals across different observational periods and that this can come with unwanted consequences, but no assessments of how much of a problem this is and how well the topic model overcomes this putative issue are performed. Further, in the discussion it is mentioned that the distribution of observed behavioral items makes the use of PCA less than ideal since one of the assumptions of PCA is normally distributed variables. This is however not the case for structural equation models. For example, the software package Mplus can model count variables (Poisson, zero inflated Poisson and negative binomial) and the flexibility of the R package OpenMx allows this as well (although involves some more programming). Taken together, the argument why the proposed machine learning algorithm improves upon more traditional methods is not backed up by a comparison with established factor analytic approaches. To help the reader assess if this new method comes with an improvement making it worthwhile to learn, it is my opinion that direct comparisons with established methods are necessary. The way I suggest this to be done is by adding up durations or number of events of individual behaviors, using exploratory and/or confirmatory factor analyses for data reduction and comparing the latent structures identified with this method to the ones from the topic model, preferably using appropriate quantitative methods. Even comparing to a method for which the data violates some assumptions could be informative to again show how big of a problem these types of violations are.

Other concerns/questions:

It seems to me that the analyses investigating the effect of covariates on the latent behaviors are based on the assumption that the underlying relationships between the observed variables and the latent states are the same for example in both males and females and across different groups. Stratified analyses would allow to investigate if the underlying structure of latent variables is the same in for example males and females, but a more important question is if the topic model allows for a structured way of testing the null hypothesis that the structures do not differ? Again, in a structural equation model framework this is straightforward by using so called “measurement invariance” models and have given important insight into sex differences in psychological constructs among other things. Can the same be done using the topic model approach? 

I have a background in using twin modelling for heritability analysis. One of the important assumptions in twin studies is the equal environment assumption, stating that the shared environment is equally similar for MZ and DZ twin pairs. It is also known that if shared environment is not modeled, this can lead to biased estimation of heritability. An equivalent to the equal environment assumption is not directly available when studying families and modeling shared environment is therefore not straightforward. It seems shared environment was not considered in the manuscript under review and I would like the authors to explain why and what potential influence on heritability this could result in.

How much of the total behavioral variance does the individual latent variables explain and how much is explained by the 10 latent states cumulatively?

There seems to be a lot of overlap regarding observed behaviors “loading” on the different latent structures. For example, the rate of for example “travel” is high for many of the behavioral states (rate above .5 for 8 out of 10 states). Should this be interpreted as “travel” being an important behavioral sub component for all of these 8 states? If so, this seems to indicate that the different states are not statistically independent, something not always desirable when using data reduction techniques. If my interpretation is correct, reporting how related the different states are to each other would be a good idea. Also, would it be possible to use the topic model approach to extract independent latent states?

Author Response

We thank the reviewer for their thoughtful and very useful comments and suggestions. We will respond point by point.

First, developing a new statistical method, or applying an established method to a new problem, using real world data can to some extent be compared to searching in the dark; it is difficult to assess if the new model captures the “real” underlying data structure since in empirical data true relationships are usually unknown. This is the reason why development of new statistical methods often includes some sort of statistical simulation, allowing data with pre-defined structures to be generated and used to evaluate the performance of the investigated statistical method. Regarding the method in the manuscript under review, I think it would for example be of interest to know how well states are identified in different scenarios like varying levels of residual error and when the true relationship between states are independent vs. non-independent.

This is of course completely true. We have included results demonstrating on simulated data that we can successfully recover latent states and phenotypes.

Second, although theoretical reasons for why the topic model could outperform more traditional methods for data reduction are mentioned in the manuscript, I think it is essential when proposing a new (and for many readers a more complicated) method that it is clearly shown how this new technique compares to established ways of addressing the same problem. Even though I agree with the authors comments about the limits of using PCA for data reduction in data like this, based on my experience with similar problems, structural equation models (including factor analytic methods) would be a better comparison. The authors mention that researchers using PCA summarize data per individuals across different observational periods and that this can come with unwanted consequences, but no assessments of how much of a problem this is and how well the topic model overcomes this putative issue are performed. Further, in the discussion it is mentioned that the distribution of observed behavioral items makes the use of PCA less than ideal since one of the assumptions of PCA is normally distributed variables. This is however not the case for structural equation models. For example, the software package Mplus can model count variables (Poisson, zero inflated Poisson and negative binomial) and the flexibility of the R package OpenMx allows this as well (although involves some more programming). Taken together, the argument why the proposed machine learning algorithm improves upon more traditional methods is not backed up by a comparison with established factor analytic approaches. To help the reader assess if this new method comes with an improvement making it worthwhile to learn, it is my opinion that direct comparisons with established methods are necessary. The way I suggest this to be done is by adding up durations or number of events of individual behaviors, using exploratory and/or confirmatory factor analyses for data reduction and comparing the latent structures identified with this method to the ones from the topic model, preferably using appropriate quantitative methods. Even comparing to a method for which the data violates some assumptions could be informative to again show how big of a problem these types of violations are.

This is of course also an excellent point. The manuscript now includes a comparison with exploratory factor analysis. The main takeaway from our perspective is that EFA, and more generally any exploratory model based on approximating the data covariance matrix such as PCA and many SQE methods, represent linear relationships between behaviors but have difficulty capturing e.g. quadratic relationships, three-way relationships, etc. The comparative advantage of the behavioral state model is that it is naturally able to capture such relationships, while also representing (in our view at least!) the underlying relationships in an intuitive way.

We have not considered more complex structural equation models that attempt to estimate, e.g., interactions and nonlinear effects between. However, our understanding (though none of the authors are experts in the SQE literature) is that those are confirmatory rather than exploratory approaches, whereas we view the behavioral state model and topic models generally as exploratory tools.

Other concerns/questions:

It seems to me that the analyses investigating the effect of covariates on the latent behaviors are based on the assumption that the underlying relationships between the observed variables and the latent states are the same for example in both males and females and across different groups. Stratified analyses would allow to investigate if the underlying structure of latent variables is the same in for example males and females, but a more important question is if the topic model allows for a structured way of testing the null hypothesis that the structures do not differ? Again, in a structural equation model framework this is straightforward by using so called “measurement invariance” models and have given important insight into sex differences in psychological constructs among other things. Can the same be done using the topic model approach?

Your impression here is correct -- sex and group membership were both included as covariates but the interaction was not. However, it is quite straightforward to build in such stratification by including the appropriate regressors. We did not include these interactions in this analysis because in the data set we were considering, we had relatively few animals outside of the largest social group F and so were sceptical of our ability to estimate such interactions.

Regarding the more general question of hypothesis testing, it is possible to do so in a manner similar to that used in classical statistics. One could fit a model with sex-by-group interactions and then compare it to the model without those interactions, e.g., the deviance information criteria (DIC), widely-applicable information criteria (WAIC), or simply cross-validation. The interpretation of these metrics is generally more about goodness-of-fit rather than null-hypothesis testing in the strict sense.

However, in this context we generally prefer to think in terms of reliable estimation of effects rather than hypothesis testing per-se. For example, given the complexity of macaque social life it would be rather surprising if being a female vs a male in one social group were exactly the same as being a female vs a male in another social group, even if such interactions were too small to definitively identify in a given data set. In the future we plan to improve the state model’s ability to handle large numbers of covariates by using shrinkage priors, which formalize the biologically-plausible notion that many covariates probably have some small effects while a few may have very large effects.

I have a background in using twin modelling for heritability analysis. One of the important assumptions in twin studies is the equal environment assumption, stating that the shared environment is equally similar for MZ and DZ twin pairs. It is also known that if shared environment is not modeled, this can lead to biased estimation of heritability. An equivalent to the equal environment assumption is not directly available when studying families and modeling shared environment is therefore not straightforward. It seems shared environment was not considered in the manuscript under review and I would like the authors to explain why and what potential influence on heritability this could result in.

This is very much a concern, and one that applies to the “animal model” literature generally. One advantage of working with the Cayo Santiago population is that many broad characteristics of the environment are invariant across animals, such as climate, general availability of food and resources, density, etc. Since these are invariant across the entire study population they will not confound estimates of genetic variation. We also attempt to measure environmental covariance among the study sample and incorporate that into the model. In the current analysis for example we model influence of the social group of which the animal is a member, as well as, depending on how broad one’s definition of environment is, dominance rank. In the past we have also included information such as the number of close kin present and maternal effects, though in our experience these have generally had minimal impacts on heritability estimates.

In general we think it is fair to say that estimates of heritability in wild or free-ranging animal populations are contingent upon how well one is able to control and measure non-genetic environmental and demographic confounds. The Cayo population is among the best in this regard, but it is always a possibility that heritability estimates will be altered by better and more detailed data that is collected in the future.

How much of the total behavioral variance does the individual latent variables explain and how much is explained by the 10 latent states cumulatively?

A rough calculation (not properly integrating over uncertainty regarding which state each observation belongs to) indicates that about 40% of the variance in the data set is explained by this model. It’s not immediately clear how to interpret the variance explained by a single state as almost definitionally, the amount of variance explained by a state will be proportional to how many observations are assigned to that state. Furthermore, states can predict more or less variable behavior, so states that predict high variability in behavior will also explain less of the variability in data.

There seems to be a lot of overlap regarding observed behaviors “loading” on the different latent structures. For example, the rate of for example “travel” is high for many of the behavioral states (rate above .5 for 8 out of 10 states). Should this be interpreted as “travel” being an important behavioral sub component for all of these 8 states? If so, this seems to indicate that the different states are not statistically independent, something not always desirable when using data reduction techniques. If my interpretation is correct, reporting how related the different states are to each other would be a good idea. Also, would it be possible to use the topic model approach to extract independent latent states?

I think the notion of independence per se is not particularly useful for thinking about latent states. The latent states can be thought of as a mixture model -- for example, imagine a bimodal gaussian distribution on two dimensions. The two modes, or “states” in our model, may be very far apart on the y-axis, but very close on the x-axis. It is less clear to us what it means for two modes to be independent than for two modes to be close or far. We agree that it is important to understand distances between states, and to what extent there might be a clustering structure among states themselves. A rigorous and systematic way of doing this is one of our future directions.

In the case with traveling, we think the reviewer’s interpretation here is correct in that traveling is ubiquitous -- in almost all contexts, macaques will do a fair amount of traveling.

Round 2

Reviewer 2 Report

I am happy to see the authors have addressed most of my concerns from the first submission including adding simulations to the study and comparing their results to factor analysis results. I still have a few questions/concerns outlined below:

The simulated data part is a very good addition to the manuscript and shows that the topic model successfully recover latent states and phenotypes. For the sake of comparison, it would however be interesting to see how well factor analysis/PCA would recover these simulated states. My point here is still that it is not completely clear to me how much better the topic model identifies latent traits than more traditional models do. I do see the theoretical benefit of the topic model, but I think with simulations set up this gives a great opportunity to direct compare this model’s performance compared to factor analysis.  

An important part of factor analyses is determining the number of factors to extract by either looking at scree plots or studying for example eigenvalues or acceleration factors. In the manuscript under review ten states were extracted but it is not clear how this number of states was determined and what was done to make sure this is the optimal number for the data. A better description of this is warranted. Related, it is concerning to me that the authors seem to be of the opinion that there is not straight forward way to get an estimate of total behavioral variance explained by each individual state, as this information could be helpful to assess the optimal number of states to extract.

My next concern has to do with interpretability. Although the authors on page 13 of the manuscript discuss possible interpretations of some of the states, it is not very clear to me when I look at figure 4 what most of the states represent. For example, S5 has a high heritability, but what ecologically relevant behavioral state is studied here? If focusing on behaviors with a relative rate above 0.5 it is not obvious to me what these 9 behaviors collectively represent. In short what would an appropriate label for S5 be, if for example incorporating all behaviors with a relative rate above 0.5? In addition, is there a meaningful difference between S4 and S5 making extracting both of these two states worthwhile? This relates to my question in the first revision about independent states and the question about if the topic model is doing a good job of data reduction when it finds for example “travel” to be an important part of all 10 states. Crossloadings, such as travel in the manuscript under review, is usually considered undesirable when using data reduction techniques such as factor analysis as it makes latent structure interpretation difficult. Taken together, if I was analyzing this data and after a first round of analyses found the 10 states identified in the manuscript I would find the “crossloading” problematic, would find the high similarity between some of the states less than ideal and would have an overall issue with state interpretability. For these reasons, I would further investigate how many states to extract and try different rotation procedures to reduce item overlap and increase interpretability of states. I think it is important for new tools to improve on old ones and more description of how the topic model can overcome some of the challenges listed in this paragraph is warranted.  

If the authors want to use heritability estimates to compare the performance of the topic model compared to factor analysis, I think the similarity between the states/factors needs to first be assessed (same thing for repeatability) to make sure the same latent structures are captured, and the data would be more easily interpreted if listed in a table so that direct comparisons can be made.   

Author Response

I am happy to see the authors have addressed most of my concerns from the first submission including adding simulations to the study and comparing their results to factor analysis results. I still have a few questions/concerns outlined below:

The simulated data part is a very good addition to the manuscript and shows that the topic model successfully recover latent states and phenotypes. For the sake of comparison, it would however be interesting to see how well factor analysis/PCA would recover these simulated states. My point here is still that it is not completely clear to me how much better the topic model identifies latent traits than more traditional models do. I do see the theoretical benefit of the topic model, but I think with simulations set up this gives a great opportunity to direct compare this model’s performance compared to factor analysis.  

While the general point is well-taken, we do not believe that fitting a factor analysis model to the simulated data would address this. The reason for this is that the data were simulated under the topic model, so it is not particularly informative to say that topic model captures the same latent states than the factor model, especially since the structure of the topic model is fundamentally different than that assumed under a topic model.

An important part of factor analyses is determining the number of factors to extract by either looking at scree plots or studying for example eigenvalues or acceleration factors. In the manuscript under review ten states were extracted but it is not clear how this number of states was determined and what was done to make sure this is the optimal number for the data. A better description of this is warranted. Related, it is concerning to me that the authors seem to be of the opinion that there is not straight forward way to get an estimate of total behavioral variance explained by each individual state, as this information could be helpful to assess the optimal number of states to extract.

The reviewer has once again identified an important omission from the manuscript. We have included information on how we chose 10 states using a goodness-of-fit measure, and also provided simulation results indicating that the measure we used performs well in the simulated data sets.

My next concern has to do with interpretability. Although the authors on page 13 of the manuscript discuss possible interpretations of some of the states, it is not very clear to me when I look at figure 4 what most of the states represent. For example, S5 has a high heritability, but what ecologically relevant behavioral state is studied here? If focusing on behaviors with a relative rate above 0.5 it is not obvious to me what these 9 behaviors collectively represent. In short what would an appropriate label for S5 be, if for example incorporating all behaviors with a relative rate above 0.5? In addition, is there a meaningful difference between S4 and S5 making extracting both of these two states worthwhile? This relates to my question in the first revision about independent states and the question about if the topic model is doing a good job of data reduction when it finds for example “travel” to be an important part of all 10 states. Crossloadings, such as travel in the manuscript under review, is usually considered undesirable when using data reduction techniques such as factor analysis as it makes latent structure interpretation difficult. Taken together, if I was analyzing this data and after a first round of analyses found the 10 states identified in the manuscript I would find the “crossloading” problematic, would find the high similarity between some of the states less than ideal and would have an overall issue with state interpretability. For these reasons, I would further investigate how many states to extract and try different rotation procedures to reduce item overlap and increase interpretability of states. I think it is important for new tools to improve on old ones and more description of how the topic model can overcome some of the challenges listed in this paragraph is warranted.  

Specifically regarding S5, an appropriate (but speculative) label might be “traveling together” -- the focal animal is often in the presence other macaques throughout the observation, with whom they are engaging in affiliative behaviors with few agonistic interactions, while traveling a fair amount. So this may be a state spent traveling with a partner or social subgroup. It is principally distinguished from S4 by SocialProximity and ProximityGroupSize -- how often the animal is in the presence of other animals and how many different animals they are around. So while certain affiliative behaviors are similarly frequent across both states, in S4 the animal is not consistently in the presence of other animals. Whether this is a worthwhile distinction is of course open for the debate and depends on the specific application under consideration.

Regarding rotation procedures in particular, those can be applied to factor analysis and similar models because they model the covariance matrix, to which rotations and other linear transformations can be applied without altering the underlying probabilistic model. This is not true of latent variable models in general -- there is no way to rotate behavioral states or topics without fundamentally altering the model itself.

We don’t consider behaviors rating similarly in different states to be a problem in and of itself because states are effectively “archetypal” behavioral observations, rather than dimensions of variation as in factor analysis. Imagine we had, for instance, measured in the ethogram the amount of time spent breathing; one would expect breathing to be consistently high across all behavioral states in a topic model, but in a factor model the ubiquity of breathing would be captured by the estimated mean amount of breathing and might not load on any particular factor. Of course, the flip side of this example is that breathing might not be an interesting behavior because it does not distinguish behavioral states from each other, and so one might choose to omit it for the sake of clarity.

We think it is more useful to think about distances between states than crossloadings. It is certainly possible that the topic model could find states that are close enough together that there are no scientifically interesting distinguishing features between them -- and one might argue that this indeed occurs in the present manuscript.  An important avenue for future work is to determine reliable ways of either biasing the model towards favoring more dissimilar states, or post-processing procedures for collapsing similar behavioral states together.

If the authors want to use heritability estimates to compare the performance of the topic model compared to factor analysis, I think the similarity between the states/factors needs to first be assessed (same thing for repeatability) to make sure the same latent structures are captured, and the data would be more easily interpreted if listed in a table so that direct comparisons can be made.

While definitionally both types of models should reflect some of the same relationships present in the data (at least to the extent that both models are capable of representing those relationships) we do not expect that factor analysis and the topic model should capture the same latent structures. The mathematical form of the models and the interpretations of the latent structures are quite different, as described above. Therefore we think it would be misleading to try to form a direct mapping between states and factors as would be suggested by a table. From our point of view the relevant metric is the distribution of repeatabilities and heritabilities of the latent structures extracted by the model, which we think is most clearly represented in plot form as in Figure 8 b&c.

In order to facilitate comparison between the behavioral states and factors, we have presented the factor loadings in a supplementary table. Since we think that direct comparisons between factor loadings and state behavior rates may be misleading and at best highly subjective for the reasons described above, we prefer to leave it to the readers to judge the extent to which they believe the states and factors do and do not match.

Round 3

Reviewer 2 Report

The authors have address enough of my concern for the paper to warrant publication in Brain Sciences.